# Energy-Dependent Endocytosis Is Involved in the Absorption of Indomethacin Nanoparticles in the Small Intestine

**DOI:** 10.3390/ijms20030476

**Published:** 2019-01-22

**Authors:** Miyu Ishii, Yuya Fukuoka, Saori Deguchi, Hiroko Otake, Tadatoshi Tanino, Noriaki Nagai

**Affiliations:** 1Faculty of Pharmacy, Kindai University, 3-4-1 Kowakae, Higashi-Osaka, Osaka 577-8502, Japan; 1833420012r@kindai.ac.jp (M.I.); 1833420015f@kindai.ac.jp (Y.F.); 1111610121m@kindai.ac.jp (S.D.); hotake@phar.kindai.ac.jp (H.O.); 2Faculty of Pharmaceutical Sciences, Tokushima Bunri University, 180 Yamashiro-Cho, Tokushima 770-8514, Japan; tanino@phar.kindai.ac.jp

**Keywords:** nanoparticle, endocytosis, oral route, indomethacin, drug delivery system

## Abstract

We previously reported that oral formulations containing indomethacin nanoparticles (IND-NPs) showed high bioavailability, and, consequently, improved therapeutic effects and reduced injury to the small intestine. However, the pathway for the transintestinal penetration of nanoparticles remained unclear. Thus, in this study, we investigated whether endocytosis was related to the penetration of IND-NPs (72.1 nm) using a transcell set with Caco-2 cells or rat intestine. Four inhibitors of various endocytosis pathways were used [nystatin, caveolae-dependent endocytosis (CavME); dynasore, clathrin-dependent endocytosis (CME); rottlerin, macropinocytosis; and cytochalasin D, phagocytosis inhibitor], and all energy-dependent endocytosis was inhibited at temperatures under 4 °C in this study. Although IND-NPs showed high transintestinal penetration, no particles were detected in the basolateral side. IND-NPs penetration was strongly prevented at temperatures under 4 °C. In experiments using pharmacological inhibitors, only CME inhibited penetration in the jejunum, while in the ileum, both CavME and CME significantly attenuated penetration. In conclusion, we found a novel pathway for the transintestinal penetration of drug nanoparticles. Our hypothesis was that nanoparticles would be taken up into the intestinal epithelium by endocytosis (CME in jejunum, CavME and CME in ileum), and dissolved and diffused in the intestine. Our findings are likely to be of significant use for the development of nanomedicines.

## 1. Introduction

[1-(4-chlorobenzoyl)-5-methoxy-2-methylindol-3-yl] acetic acid, indomethacin, is a non-steroidal anti-inflammatory drug (NSAID), one of the most widely prescribed groups of drugs for the treatment of inflammation and pain. However, the long-term use of NSAIDs is well-known to cause severe ulceration and inflammation of the small intestine [1]. Many factors, such as nitric oxide (NO), prostaglandin (PG), and intestinal bacteria, are known to be involved in the mechanisms of NSAID-induced small-intestinal injury [2,3,4,5]. NSAIDs hamper the synthesis of PGE2 by inhibiting cyclooxygenase (COX), thereby reducing mucus secretion and increasing small-intestinal motility [6], resulting in minute injuries that allow intestinal bacteria to invade the mucosa. The production of inducible NO synthase caused by the bacteria leads to the production of reactive oxygen species and NO. These gases induce neutrophil activation and thereby COX-2 induction, as well as the production of inflammatory cytokines and mediators, which results in the onset of inflammation in the small intestine [7]. Thus, the erosions and ulcers within the small intestine are caused by an NSAID-impaired mucosal defense system [8,9]. On the other hand, it may be efficient to enhance drug absorption to prevent the side effects of indomethacin, since reducing the dose would prevent stimulation by the drugs and the inhibition of COX in local tissues (e.g., the small intestine).

As formulation technology has advanced, prodrugs have been used as a method to reduce the side effects and improve the bioavailability (BA) of indomethacin [10,11,12]. In addition, the successful introduction of nanotechnology has brought more oral nanomedicines onto the market [13], and improved the therapeutic efficacy and/or BA of drugs [14]. In a previous study, we designed a solid nanoparticle formulation containing indomethacin (IND-NPs), and showed that the transintestinal penetration of IND-NPs was higher than that of dissolved indomethacin or indomethacin microparticles [15]. In addition, the BA of IND-NPs was approximately five-fold that of commercially available indomethacin preparations (microparticles), making it possible to avoid the side effects (small-intestinal injury) of indomethacin in rats by reducing its dose [15]. Therefore, IND-NPs may resolve the issue of NSAID-induced small-intestinal injury. However, the mechanism for the transport of drug nanoparticles through the intestine is not fully understood, and it is important to elucidate the pathway for the transintestinal penetration of IND-NPs.

Recently, it was reported that energy-dependent endocytosis is related to the trafficking pathways of drug nanoparticles [16,17]. Endocytosis pathways are mainly classified as clathrin-dependent endocytosis (CME), caveolae-dependent endocytosis (CavME), macropinocytosis (MP), or phagocytosis [18,19]. In this study, we investigated whether the uptake of IND-NPs was related to these trafficking pathways to elucidate the mechanism of transintestinal penetration in the oral administration of drug nanoparticles.

## 2. Results

### 2.1. Design of Oral Formulation Containing Indomethacin Nanoparticles

Firstly, we prepared indomethacin nanoparticles by a bead mill method, and obtained an overview of the particle size distribution by using a laser diffraction particle size analyzer, SALD-7100. The mean particle size in the indomethacin powder (microparticles) was 16.3 ± 12.9 μm (Figure 1A). The particle size was decreased by bead mill treatment so that the mean particle size in the IND-NPs was 74 ± 52 nm (Figure 1B). Next, we demonstrated the accurate size and shape of the indomethacin nanoparticles using a dynamic light-scattering NANOSIGHT LM10 and atomic force microscope (AFM) imaging, respectively. The AFM imaging showed the indomethacin after bead mill treatment to be in the nano-order (Figure 1D) with a mean particle size in the IND-NPs (nanoparticles) of 72.1 ± 4.3 nm (Figure 1C). In these formulations, the indomethacin consisted of two types (solution type and solid type), and it was important to investigate the ratio of solution/solid type in the IND-NPs. Figure 1E shows the changes in solubility of indomethacin before and after the bead mill treatment. The drug solubility of the IND-NPs was 131.2 μM, which is 10-fold that of the IND-MPs (25.7 μM). The ratio of dissolved indomethacin was 9.4% in the IND-NPs formulation (0.05%).

### 2.2. Stability of the Oral Formulation Containing Indomethacin Nanoparticles

It is known that nanoparticles without suitable additives aggregate easily. In evaluating the stability in the formulation used, it was important to elucidate the transport pathway of nanoparticles. Therefore, we investigated whether the particle size frequency and shape of the indomethacin in IND-NPs had changed 30 days after preparation. Figure 2A,B show the particle size frequencies (Figure 2A) and AFM image (Figure 2B) 30 days after preparation. The particle size of the indomethacin remained in the nano-size order at 84.3 ± 4.9 nm. In addition, no degradation (Figure 2C) or decrease in the number (Figure 2D) of indomethacin nanoparticles in IND-NPs were observed after 30 days. Furthermore, there was no detectable precipitation or aggregation (Figure 2E,F).

### 2.3. Effect of the Energy-Dependent Endocytosis on the Transintestinal Penetration of Indomethacin Nanoparticles Using Caco-2 Cell Monolayers

Some researchers have reported energy-dependent endocytosis to be related to the penetration of nanoparticles into the cell [20]; however, there is no report in which the role of endocytosis in the transintestinal penetration of solid nanoparticles was investigated. Therefore, we demonstrated the changes in the transintestinal penetration of indomethacin nanoparticles in human epithelial colorectal adenocarcinoma (Caco-2) cell line monolayers inhibited for various types of energy dependent endocytosis. Energy-dependent endocytosis was inhibited by incubation at 4 °C, with TER over 400 Ω·cm^2^ at 60 min (Figure 3A). Under normal conditions (37 °C), the accumulation of indomethacin from IND-NPs was greater than from IND-MPs, and tended to be low in comparison with an IND solution (Figure 3B). In addition, the penetration of indomethacin from IND-NPs was greater than that from both IND-MPs and IND solution (Figure 3C), and indomethacin nanoparticles were detected in the basolateral side at 4 °C (Figure 3D,E). On the other hand, the accumulation and penetration of indomethacin nanoparticles were significantly decreased (Figure 3B,C), and no indomethacin particles were observed in the basolateral side at 4 °C (Figure 3D). In this study, we investigated the effect of endocytosis inhibitors on the accumulation and penetration of indomethacin nanoparticles in Caco-2 cell monolayers (Figure 4). During the sampling period, the TER values of the Caco-2 cell monolayers treated with nystatin (CavME inhibitor), dynasore (CME inhibitor), rottlerin (MP inhibitor), and cytochalasin D (phagocytosis inhibitor) were 381 ± 20 Ω·cm^2^, 375 ± 23 Ω·cm^2^, 392 ± 17 Ω·cm^2^, and 391 ± 18 Ω·cm^2^, respectively, while the TER was 367 ± 31 Ω·cm^2^ in the group co-treated with nystatin and dynasore. Nystatin tends to prevent the accumulation and penetration of indomethacin nanoparticles, and dynasore significantly attenuates the accumulation and penetration of indomethacin nanoparticles in the Caco-2 cell monolayers. The accumulation, penetration, and particle number in the groups treated with dynasore were 79.3%, 67.0%, and 67.0% of the vehicle, respectively (Figure 4A,C,E). Figure 4B,D,F shows the changes in the accumulation (Figure 4A), penetration (Figure 4D) and particle number (Figure 4F) of indomethacin nanoparticles in Caco-2 cell monolayers co-treated with nystatin and dynasore. Both the accumulation and penetration of indomethacin nanoparticles were strongly inhibited by this co-treatment with values of 75.9%, 63.0%, and 55.5% of the vehicle, respectively.

### 2.4. Effect of Energy-Dependent Endocytosis on the Transintestinal Penetration of Indomethacin Nanoparticles in the Rat Jejunum and Ileum

Figure 5 and Figure 6 show the changes in the penetration of indomethacin nanoparticles in the jejunum (Figure 5) and ileum (Figure 6); Table 1 and Table 2 summarize the pharmacokinetic parameters calculated from the transintestinal penetration data in the jejunum (Table 1) and ileum (Table 2). The transintestinal penetration of indomethacin nanoparticles in both jejunum and ileum was prevented at 4 °C and by co-treatment with nystatin and dynasore, with no indomethacin particles observed in the basolateral at 4 °C. Although dynasore significantly inhibited the area under the drug concentration-time curve in the reservoir chamber (AUC) in the jejunum, the AUC values in jejunum treated with other endocytosis inhibitors (nystatin, rottlerin, or cytochalasin D) were similar to that of the vehicle (Figure 5C,D). In addition, the *J*_c_ in the jejunum treated with dynasore was also lower in comparison with the vehicle-, nystatin-, rottlerin-, and cytochalasin D-treated groups (Table 1). On the other hand, the AUC and *J*_c_ in the ileum treated with nystatin and dynasore were significantly lower than in the vehicle (Table 2, Figure 6C,D). In particular, dynasore strongly attenuated the penetration of indomethacin nanoparticles in both jejunum and ileum.

## 3. Discussion

Nanomedicines are recognized to have significantly improved cellular uptake and blood circulation time as compared to traditional formulations [20], and nanoparticle-based drug delivery systems have been introduced in many studies. Silica- and biodegradable-nanoparticles, dendrimers, polymer-drug conjugates, liposomes, quantum dots, polymeric micelles, etc. are classic examples of nanoparticulate materials currently being studied in laboratories, and some formulation technologies are already in use clinically [21]. We also designed a method to prepare solid nanoparticles using the bead mill method and additives, such as methylcellulose (MC) and 2-hydroxypropyl-β-cyclodextrin (HPβCD) [15,22,23,24,25], and reported that IND-NPs (an oral formulation containing solid indomethacin nanoparticles) shows a high BA that improves its therapeutic effect and reduces side effects, such as small-intestinal injury [15]. However, the pathway for the transintestinal penetration of solid nanoparticles is still unclear. In this study, we found that the high BA of indomethacin nanoparticles is led by a combination of CavME and CME and, in particular, the CME pathway is strongly involved in drug absorption in the both jejunum and ileum.

Oral administration has been widely applied clinically, since the oral route shows simple operability and excellent patient compliance, and the method is considered to be convenient and the most suitable for chronic therapies. Firstly, we prepared the IND-NPs, and evaluated its stability. The particle size of indomethacin remained in the nano-size order (Figure 1), and there was no degradation or change in the number of indomethacin nanoparticles in IND-NPs was for 30 days (Figure 2). Further, no precipitation or aggregation was observed in the IND-NPs for 30 days after preparation (Figure 2). These results show that it is possible to prepare an oral formulation containing 0.05% indomethacin nanoparticles.

In this study, we investigated whether the drug nanoparticles were endocytosed in the intestine, and whether endocytosis was related to the high transintestinal penetration of IND-NPs. In the experiments on the inhibition of endocytosis, cold-temperature incubation and selective pharmacological inhibitors were used. Previous studies showed that all energy-dependent uptake into cells was inhibited under cold-temperature incubation (4 °C) [26], and it is known that different endocytosis pathways, CavME, CME, MP, and phagocytosis, are inhibited by treatment of nystatin [27], dynasore [28], rottlerin [29], and cytochalasin D [27], respectively. On the other hand, drug penetration from the small intestine into the blood requires a drug to cross at least two barriers, the mucus layer and epithelial cells. The mucus layer is comprised mainly of mucin protein, and forms the first barrier between the small intestine and epithelial cells [30], and the epithelial cell layer is reinforced by tight junctions between the cells, as is the second barrier [30,31]. Therefore, in this study, the transintestinal penetration of IND-NPs was demonstrated using two different approaches: an in vitro permeation test using Caco-2 cell monolayers, and an ex vivo study using excised rat jejunum and ileum. In the experiment using Caco-2 cell monolayers, indomethacin in IND-NPs was penetrated into the basolateral side at 37 °C, although no indomethacin particles were observed in the basolateral side at 4 °C (Figure 3). In addition, the accumulation and penetration of indomethacin in IND-NPs were prevented under 4 °C conditions (Figure 3), as well as by dynasore in the Caco-2 cell monolayers. Both the accumulation and penetration of indomethacin nanoparticles were strongly inhibited by co-treatment with nystatin and dynasore (Figure 4). These results suggest that indomethacin nanoparticles can penetrate the intestinal epithelial cell monolayers as nanoparticles, and the CME was related to the acceleration of the drug penetration via cell monolayers.

Next, we demonstrated the involvement of endocytosis pathways in drug penetration from IND-NPs in an ex vivo study using excised rat jejunum and ileum. As in the case using Caco-2 cell monolayers, the transintestinal penetration of indomethacin nanoparticles was inhibited at 4 °C in both jejunum and ileum, and the indomethacin nanoparticles themselves did not penetrate through the intestine at either 4 °C or 37 °C. We then used the inhibitors to elucidate the endocytosis pathway related to the transintestinal penetration of indomethacin in IND-NPs. In the jejunum, only dynasore inhibited the penetration of indomethacin in IND-NPs. These data are similar to the results for Caco-2 cell monolayers. On the other hand, penetration was prevented by both nystatin and dynasore in the ileum. It is known that the particle sizes corresponding to the CavME, CME, and MP pathways are <80 nm, <120 nm, and 100 nm–5 μm, respectively [32]. Since the size of indomethacin nanoparticles in IND-NPs is approximately 50–200 nm (mean particle size 72.1 nm, Figure 1C), this suggests a suitable size for the CavME and CME pathways. In addition, it has been reported that the percentage of transcytosis of nanoparticles mediated by CavME and CME is higher than that mediated by MP in previous studies using intestinal epithelial cell lines (Caco-2 and HT-29) [33]. Taken together, it is suggested that the activity of each endocytosis pathway is different in the jejunum and ileum, and that drug nanoparticles mainly penetrate by the CME pathway in the jejunum. On the other hand, the CavME and CME pathways are strongly involved in the penetration of drug nanoparticles in the ileum. These results support the previous studies of Yang et al. [33]. Further studies are needed to clarify the novel drug delivery pathway of solid nanoparticles via energy-dependent endocytosis in the small intestine. Therefore, we are now planning to demonstrate the correlation between drug particle size and the activation of endocytosis pathways using an immunohistochemical approach.

## 4. Materials and Methods 

### 4.1. Animals

Male 7-week-old Wistar rats (approximately 220 g) were obtained from Kiwa Laboratory Animals Co., Ltd. (Wakayama, Japan), and were provided a CE-2 formulation diet (Clea Japan Inc., Tokyo, Japan) and water, and housed under the following conditions: 25 °C, and exposed to light from 7:00 a.m. to 7:00 p.m., and at other times dark. All experiments were performed in accordance with the Pharmacy Committee Guidelines for the Care and Use of Laboratory Animals in Kindai University and the Guiding Principles approved by the Japanese Pharmacological Society, and the guidelines for animal experimentation of the International Association for the Study. The experiments using rats were approved on 1 April 2013 (project identification code KAPS-25-004) by Kindai University.

### 4.2. Chemicals

Dulbecco’s Modified Eagle’s Medium (high-glucose, DMEM), non-essential amino acids solution, rottlerin, and dynasore were purchased from Nacalai Tesque (Kyoto, Japan), and heat-inactivated fetal bovine serum, penicillin, and streptomycin were obtained from Grand Island Biological Company (GIBCO, Tokyo, Japan). l-glutamine and nystatin were purchased from Sigma-Aldrich Japan (Tokyo, Japan). Indomethacin powder (particle size, 16.3 ± 12.9 μm), 4-(2-hydroxyethyl)-1-piperazineethanesulfonic acid (HEPES) buffer, propyl p-hydroxybenzoate, isoflurane, and cytochalasin D were provided by Wako Pure Chemical Industries, Ltd. (Osaka, Japan). HPβCD and MC (type SM-4) were supplied by Nihon Shokuhin Kako Co., Ltd. (Tokyo, Japan) and Shin-Etsu Chemical Co., Ltd. (Tokyo, Japan), respectively. Pentobarbital was obtained from Tokyo Chemical Industry Co., Ltd (Tokyo, Japan), and methanol and acetonitrile were provided by Kanto Chemical Co., Inc. (Tokyo, Japan). The Bio-Rad Protein Assay Kit was purchased from Bio-Rad Laboratories, Inc. (Bio-Rad, CA, U.S.A.). All other chemicals were of the highest purity which was commercially available.

### 4.3. Preparation of IND-NPs

IND-NPs (formulation containing indomethacin nanoparticles) was prepared following our previous report [15]. Briefly, indomethacin powder (microparticles) was added into MC powder, and mixed at 3000 rpm for 30 s at 4 °C using 1 mm zirconia beads and Bead Smash 12 (a bead mill, Wakenyaku Co. Ltd, Kyoto, Japan). After that, the mixture was added into HPβCD solution, and crushed with the 0.1 mm zirconia beads and Bead Smash 12 (IND-NPs). The conditions for the bead mill were as follows: 5500 rpm for 30 s × 30 times at 4 °C. IND-MPs (formulation containing indomethacin microparticles) was prepared by mixing indomethacin powder, MC and HPβCD. The compositions of IND-MPs and IND-NPs were as follows: 0.05% indomethacin, 0.5% MC, and 5% HPβCD. 0.05% IND-NPs is equivalent to 1.4 mM IND, and the pH was 6.5. In this penetration process, the combination of MC and the bead mill method was used by decreasing the IND particle size, and the cohesion of nanoparticulate solids adsorption was prevented by the surface of HPβCD. The indomethacin concentrations in the samples were measured by an HPLC method using a LC-20AT system (HPLC, Shimadzu Corp., Kyoto, Japan). An Inertsil^®^ ODS-3 column (2.1 mm × 50 mm, GL Science Co., Inc., Tokyo, Japan) was used at 35 °C, and 1 μg/mL propyl *p*-hydroxybenzoate was selected as an internal standard. The mobile phase consisted of acetonitrile/50 mM acetic acid (40/60, *v*/*v*%) at a flow rate of 0.25 mL/min, and the wavelength for detection was 254 nm [15]. In addition, the ratio of solution/solid type in the IND-MP and IND-NPs were measured by using 25 nm pore-size membrane filters (MF™-MEMBRANE FILTER, Merck Millipore, Tokyo, Japan).

### 4.4. Analysis of Particle Size and Number of Indomethacin Nanoparticles

The size and number of indomethacin nanoparticles were measured by both a laser diffraction particle-size analyzer SALD-7100 (Shimadzu Corp., Kyoto, Japan) and dynamic light-scattering NANOSIGHT LM10 (QuantumDesign Japan, Tokyo, Japan), and the indomethacin nanoparticles images were provided by an SPM-9700 (Shimadzu Corp., Kyoto, Japan). The conditions for the NANOSIGHT LM10 were as follows: measurement time, 60 s; wavelength, 405 nm (blue); viscosity of the suspension; 1.27 mPa∙s. The refractive index was set at 1.60-0.010i in the SALD-7100, and the AFM indomethacin nanoparticle image was created by a combination of phase and height images.

### 4.5. Evaluation of Dispersibility in IND-NPs

The experiment was performed according to our previous report [34]. Briefly, IND-NPs samples were stored in the dark at 20 °C for 30 days, and the changes in concentration, particle size, and nanoparticle number were measured as an evaluation of the dispersibility of IND-NPs. The concentration, particle size, and nanoparticle number of indomethacin were measured by the HPLC methods and NANOSIGHT LM10, as described above.

### 4.6. Measurement of Indomethacin Penetration through Caco-2 Cell Monolayers

Caco-2 cells were cultured in DMEM containing 10% heat-inactivated fetal bovine serum, 100 mg/mL streptomycin, 100 IU/ml penicillin, 1% l-glutamine, and 1% non-essential amino acid solution (*v*/*v*%). The Caco-2 cells (3.5 × 10^5^ cells/well) were seeded onto Transwell-ClearTM plates (Costar, Cambridge, MA, USA), and incubated for 20 days under standard conditions (5% CO_2_, 37 °C). The groups showing transepithelial electrical resistance (TER) values greater than 450 Ω·cm^2^ were used for permeation studies, and the TER was measured using an epithelial Volt-Ohm meter (Millicell-ERS, Millipore Co., Bedford, MA, USA). In experiments used to evaluate the accumulation and permeation of indomethacin, the medium was removed, and the 1.5 mL Hank’s balanced salt solution (pH 7.4) containing 25 mM HEPES (HBSS/HEPES solution) was added to the basolateral side (reservoir chamber) of the monolayer. Then, 0.5 mL vehicle or 0.05% IND-NPs was added to the apical side (donor chamber). The penetration of indomethacin was evaluated by measuring the indomethacin concentration in sample solutions (10 μL) withdrawn from the reservoir chamber. TER was measured for 0–60 min under cold (4 °C) and normal (37 °C) conditions, and the samples to determine penetration from were withdrawn at 60 min. In addition, the state of the penetrated indomethacin was analyzed based on the number of nanoparticles in the reservoir chamber. The accumulation of indomethacin was determined in cells collected 60 min after treatment, and indicated as pmol/mg protein. The protein levels were determined using a Bio-Rad Protein Assay Kit, and the indomethacin concentration, particle size, and nanoparticle number by the HPLC methods and NANOSIGHT LM10 described above.

### 4.7. Measurement of Indomethacin Penetration through Removed Small Intestine in Rats

Rats were killed by injecting a lethal dose of pentobarbital, and the small intestines were removed. The excised intestine was divided into the jejunum (29.0 ± 4.3 cm, 40% of the upper part), and ileum (43.2 ± 7.8 cm, 60% of the lower part) (mean ± SE, *n* = 13). These jejunum and ileum samples were washed in saline, and set on a methacrylate cell designed for small-intestinal penetration experiments. The donor chamber (apical side) was filled with IND-NPs, and the other side of the chamber (reservoir chamber) was filled with HEPES buffer [K_2_HPO_4_ (1 mM), glucose (5.5 mM), HEPES (10 mM), KCl (5.3 mM), NaCl (136.2 mM), and CaCl_2_ (1.7 mM), pH 7.4]. The experiments were performed at either 4 °C or 37 °C for 6 h, and the area under the drug concentration-time curve in the reservoir chamber (AUC) was determined according to the trapezoidal rule up to the last indomethacin concentration measurement point (6 h). The indomethacin concentrations were determined by the HPLC method described above, and were evaluated by the following Equations (1)–(3) [34]:
(1)$$J_{c}=\frac{K_{c}\cdot D\cdot C_{\mathrm{IND}}}{\delta }=K_{p}\cdot C_{\mathrm{IND}}$$
(2)$$\tau =\frac{\delta^{2}}{6D}$$
(3)$$Q_{t}=\frac{K_{c}\cdot D\cdot C_{\mathrm{IND}}}{\delta }=J_{c}\cdot A\cdot \left(t-\tau \right)$$
where the indomethacin content in the oral formulation, lag time, thickness of the small intestine, total amount of indomethacin appearing in the reservoir solution at time *t*, effective area of the small intestine, diffusion constant within the small intestine, indomethacin penetration rate, small intestine/preparation partition coefficient, and penetration coefficient through the small intestine are expressed as *C*_IND_, τ, *δ*, *Q*_t_, *A*, *D*, *J*_c_, *K*_m_, and *K*_p_, respectively.

### 4.8. Inhibition of Energy-Dependent Endocytosis

In the evaluation of the correlation between indomethacin nanoparticle absorption and energy-dependent endocytosis, the Caco-2 cell monolayers and removed rat small intestine were thermoregulated at 4 °C, under which condition energy-dependent endocytosis is inhibited [26], or at 37 °C (normal conditions). In addition, 54 μM nystatin (CavME inhibitor) [27], 40 μM dynasore (CME inhibitor) [28], 2 μM rottlerin (MP inhibitor) [29], or 10 μM cytochalasin D (phagocytosis inhibitor) [27] were used for the analysis of the different endocytosis pathways by pharmacological inhibitors. The endocytosis inhibitors were dissolved in 0.5% DMSO (vehicle). In the experiments using Caco-2 cell monolayers and removed rat small intestine, the endocytosis inhibitors were added 5 min prior to treatment with IND-NPs, and the reservoir chamber (methacrylate cell) in the ex vivo study using excised rat jejunum and ileum was filled with HEPES buffer containing vehicle or endocytosis inhibitors. Samples for the penetration evaluation were obtained 60 min after treatment with IND-NPs.

### 4.9. Statistical Analysis

Student’s *t*-test was used for two group comparisons, and one-way analysis of variance (ANOVA), followed by Dunnett’s multiple comparison, was used for multiple group comparisons. A minimum *P* value of 0.05 was chosen as the significance level (*P* < 0.05). The sample numbers (*n*) are shown in the Figure legends. The data from the laser diffraction particle-size analyzer (SALD-7100) are expressed as the mean ± standard deviation (SD); other data are expressed as the mean ± standard error (SE) of the mean. In the evaluation of particle sizes, the SD in SALD-7100 reflects the particle-size distribution (range, https://www.an.shimadzu.co.jp/powder/lecture/practice/p01/lesson20.htm. Accessed on 13 Jan. 2019), and the SE in NANOSIGHT LM10 shows the deviation of mean particle sizes.

## 5. Conclusions

In this study, we succeeded in preparing IND-NPs (an oral formulation containing indomethacin nanoparticles), and found a novel pathway of transintestinal penetration for drug solid nanoparticles. It was hypothesized that indomethacin nanoparticles would be taken up into the intestinal epithelium by energy-dependent endocytosis (especially CME in the jejunum, and CavME and CME in the ileum), and that the penetrated indomethacin nanoparticles would dissolve and diffuse in the intestine, since only dissolved indomethacin was observed in the reservoir chamber (pathway 2, Figure 7). As another pathway (pathway 1), dissolved indomethacin in the formulation may also pass into the small intestine, because the solubility of indomethacin in IND-NPs is 5.1-fold that in IND-MPs (Figure 1E), and indomethacin was also detected in the in vitro permeation test using Caco-2 cell monolayers and the ex vivo study using excised rat jejunum and ileum with cold or inhibitor treatment (Figure 3, Figure 4, Figure 5 and Figure 6). These findings provide significant information that can be used to design further studies aimed at developing oral nanomedicines.

## Figures and Tables

**Figure 1 ijms-20-00476-f001:**
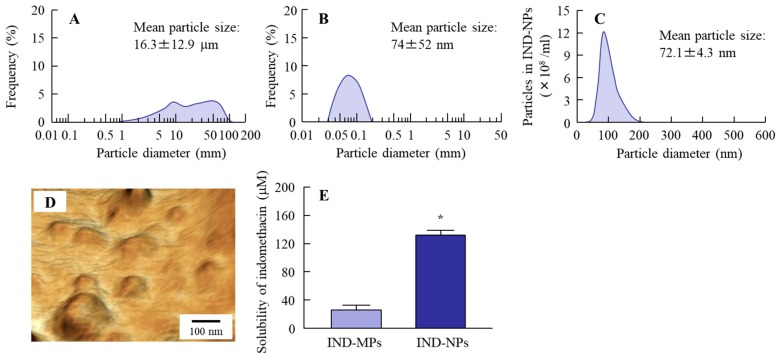
Changes in particle size frequencies and solubility of indomethacin in indomethacin nanoparticles (IND-NPs). (**A**) Particle size frequencies of indomethacin in IND-MPs; (**B**–**E**) Particle size frequencies (**B**,**C**), atomic force microscope (AFM) image (**D**), and solubility (**E**) of indomethacin in IND-NPs. Data of Figure 1A,B were obtained by a laser diffraction particle size analyzer, and Figure 1C was measured using the dynamic light-scattering method. Figure 1D was observed by SPM-9700. *n* = 5. **P* < 0.05 vs. IND-MPs. The particle size of indomethacin in IND-NPs was 50–200 nm, and 90.6% of indomethacin in the IND-NPs was of solid type (not dissolved type).

**Figure 2 ijms-20-00476-f002:**
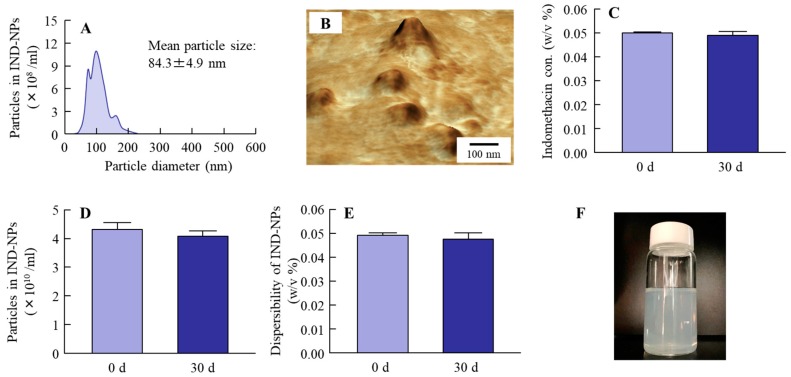
Changes in the stability of IND-NPs 30 days after bead mill treatment. (**A**–**F**); Particle size frequencies (**A**), AFM image (**B**), concentration (**C**), particle number (**D**), dispersibility (**E**), and photograph (**F**) of indomethacin in IND-NPs. The particle size and particle number were measured by the dynamic light scattering method, and the AFM image was obtained using SPM-9700. *n* = 5. The IND-NPs remained stable 30 days after preparation.

**Figure 3 ijms-20-00476-f003:**
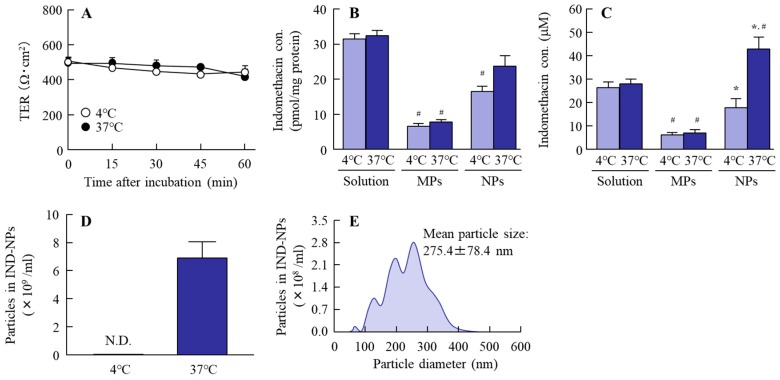
Penetration of indomethacin in IND-NPs at 4 and 37 °C through Caco-2 cell monolayers. (**A**) Changes in transepithelial electrical resistance (TER) of Caco-2 cell monolayers treated with IND-NPs. (**B**,**C**) Accumulation (**B**) and penetration (**C**) of indomethacin through Caco-2 cell monolayers treated with IND-solutions, IND-MPs, and IND-NPs. (**D**,**F**) Number (**D**) and size frequencies (**E**) of indomethacin nanoparticles in the basolateral side under at 4 °C. *n* = 6. **P* < 0.05 vs. 37 °C for each category. ^#^*P* < 0.05 vs. IND-solution at 37 °C. The accumulation and penetration of indomethacin nanoparticles were significantly decreased at 4 °C.

**Figure 4 ijms-20-00476-f004:**
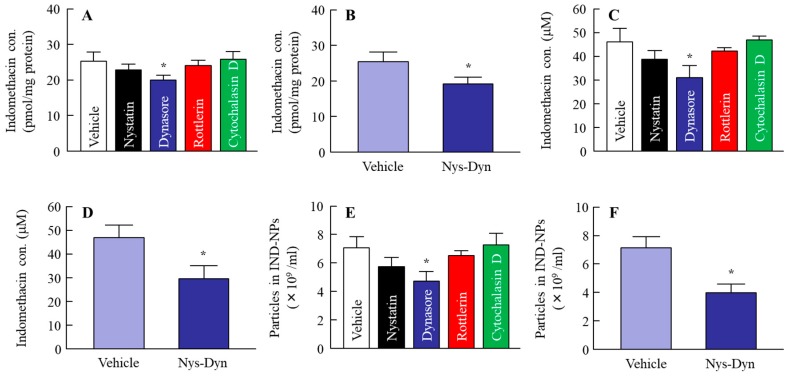
Penetration of IND-NPs through Caco-2 cell monolayers treated with endocytosis inhibitors. (**A**,**B**) Accumulation of indomethacin in the Caco-2 cell monolayers. (**C**,**D**) Penetration of indomethacin through Caco-2 cell monolayers. (**E**,**F**) Number of indomethacin nanoparticles in the basolateral side of Caco-2 cell monolayers. Nys-Dyn indicates co-treatment with nystatin and dynasore. *n* = 5–9. **P* < 0.05 vs. vehicle for each category. Dynasore significantly attenuated the accumulation and penetration of indomethacin nanoparticles in Caco-2 cell monolayers.

**Figure 5 ijms-20-00476-f005:**
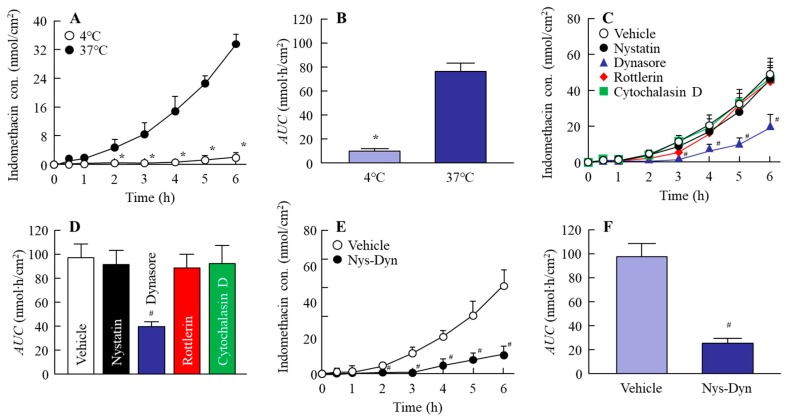
Endocytosis enhances the penetration of indomethacin nanoparticles in the jejunum. (**A**,**B**) Changes in the penetration profile (**A**) and area under the drug concentration-time curve in the reservoir chamber (AUC) (**B**) of indomethacin nanoparticles in the jejunum under 4 °C or 37 °C conditions. (**C**–**F**) Changes in the penetration profile (**C**,**E**) and AUC (**D**,**F**) of indomethacin nanoparticles in jejunum treated with endocytosis inhibitors. Nys-Dyn indicates co-treatment with nystatin and dynasore. *n* = 5–8. **P* < 0.05 vs. 37 °C conditions. ^#^*P* < 0.05 vs. vehicle for each category. Dynasore significantly attenuated the penetration of indomethacin nanoparticles in the jejunum.

**Figure 6 ijms-20-00476-f006:**
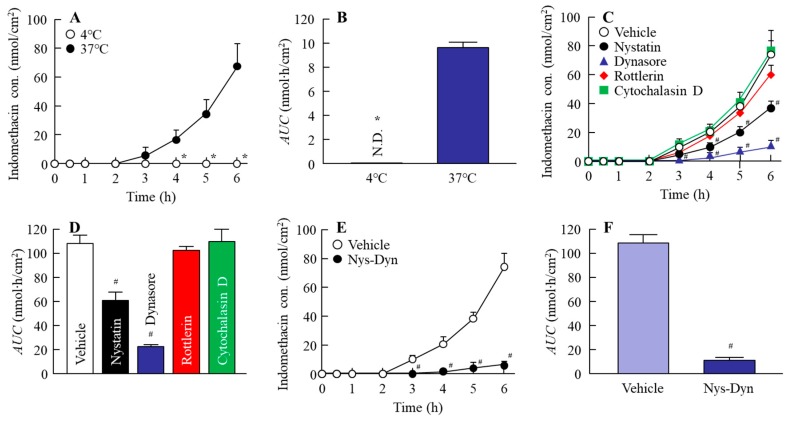
Endocytosis enhances the penetration of indomethacin nanoparticles in the ileum. (**A**,**B**) Changes in penetration profile (**A**) and AUC (**B**) of indomethacin nanoparticles in the ileum under 4 °C or 37 °C conditions. (**C**–**F**) Changes in the penetration profile (**C**,**E**) and AUC (**D**,**F**) of indomethacin nanoparticles in ileum treated with endocytosis inhibitors. Nys-Dyn indicates co-treatment with nystatin and dynasore. *n* = 5–8. **P* < 0.05 vs. 37 °C conditions. ^#^*P* < 0.05 vs. vehicle for each category. Both of nystatin and dynasore significantly attenuated the penetration of indomethacin nanoparticles in the ileum.

**Figure 7 ijms-20-00476-f007:**
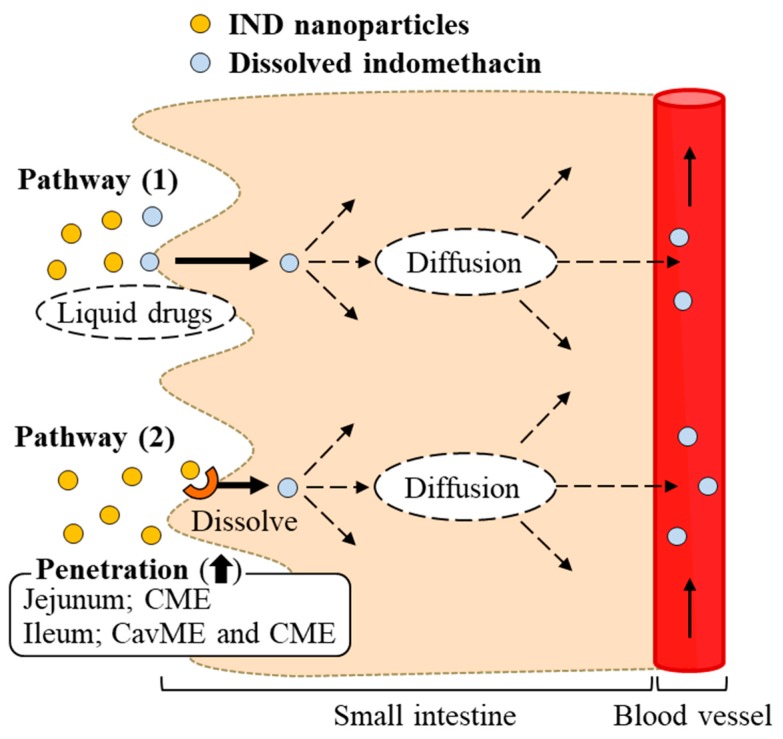
Scheme for intestinal absorption following the oral administration of IND-NPs formulation.

**Table 1 ijms-20-00476-t001:** Pharmacokinetic parameters for IND-NPs penetration in jejunum treated under cold (4 °C) temperature conditions or with endocytosis inhibitors.

Treatment	*J*_c_ (pmol/cm^2^/h)	*K*_p_ (×10^−5^/h)	*K*_m_ (×10^−3^)	*τ* (h)	*D* (×10^−3^ cm^2^/h)
Normal (37 °C treatment)	2.6 ± 0.6	5.0 ± 1.1	1.6 ± 0.4	0.69 ± 0.12	2.7 ± 0.6
4 °C treatment	0.2 ± 0.1*^,#^	0.3 ± 0.1*^,#^	0.2 ± 0.1*^,#^	1.16 ± 0.18*^,#^	1.4 ± 0.3*^,#^
Vehicle	3.9 ± 1.0	7.5 ± 1.5	2.6 ± 0.3	0.72 ± 0.11	2.5 ± 0.4
Nystatin	3.6 ± 0.9	6.9 ± 1.3	2.4 ± 0.3	0.72 ± 0.10	2.6 ± 0.4
Dynasore	1.9 ± 0.1*^,#^	3.9 ± 0.4*^,#^	1.9 ± 0.6*^,#^	0.85 ± 0.15	2.0 ± 0.7
Rottlerin	3.8 ± 1.0	7.1 ± 1.1	2.5 ± 0.4	0.70 ± 0.13	2.3 ± 0.4
Cytochalasin D	3.9 ± 1.2	7.3 ± 1.3	2.6 ± 0.5	0.71 ± 0.11	2.3 ± 0.4
Nys-Dyn	0.7 ± 0.2*^,#^	1.2 ± 0.4*^,#^	0.5 ± 0.1*^,#^	0.71 ± 0.27	2.3 ± 0.7

The data in Figure 5A,C,E were calculated according to Equations (1)–(3). *n* = 5–8. ^*^*P* < 0.05, vs. Normal for each category. ^#^*P* < 0.05, vs. Vehicle for each category.

**Table 2 ijms-20-00476-t002:** Pharmacokinetic parameters for IND-NPs penetration in the ileum treated under cold (4 °C) temperature conditions or with endocytosis inhibitors.

Treatment	*J*_c_ (pmol/cm^2^/h)	*K*_p_ (×10^−4^/h)	*K*_m_ (×10^−3^)	*τ* (h)	*D* (×10^−3^ cm^2^/h)
Normal (37 °C treatment)	7.4 ± 1.1	1.3 ± 0.2	8.9 ± 1.3	1.21 ± 0.14	1.4 ± 0.1
4 °C treatment	–	–	–	–	–
Vehicle	7.5 ± 1.0	1.4 ± 0.3	9.0 ± 0.9	1.18 ± 0.11	1.5 ± 0.2
Nystatin	5.0 ± 1.4*^,#^	0.9 ± 0.2*^,#^	8.1 ± 1.8	1.65 ± 0.37	1.3 ± 0.4
Dynasore	2.4 ± 0.1*^,#^	0.5 ± 0.1*^,#^	2.6 ± 0.7*^,#^	0.95 ± 0.21	1.9 ± 0.4
Rottlerin	6.8 ± 0.7	1.0 ± 0.2	8.7 ± 1.0	1.09 ± 0.11	1.5 ± 0.4
Cytochalasin D	7.7 ± 1.4	1.4 ± 0.3	9.1 ± 0.9	1.14 ± 0.13	1.5 ± 0.3
Nys-Dyn	1.4 ± 0.4*^,#^	0.3 ± 0.1*^,#^	1.9 ± 0.4*^,#^	0.92 ± 0.20	1.8 ± 0.3

The data in Figure 6A,C,E were calculated according to Equations (1)–(3). n = 5–8. ^*^*P* < 0.05, vs. Normal for each category. ^#^*P* < 0.05, vs. Vehicle for each category.

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
