# Peer review of "Energy-Dependent Endocytosis Is Involved in the Absorption of Indomethacin Nanoparticles in the Small Intestine"

_ijms, 2019, doi:10.3390/ijms20030476_

Round 1
Reviewer 1 Report
The manuscript describes the mechanism of absorption of indomethacin-containing nanoparticles in the small intensine. In my opinion, presented reasearch is interesting and the subject is worth investigation. The article is cosistent and well written and can be accepted for publication in Int. J. Mol. Sci. in the present form.
Author Response
We carefully revised our manuscript according to the suggestions of the reviewer 1, and details are as follows.
<Q and A for Reviewer 1>
Q1. The manuscript describes the mechanism of absorption of indomethacin-containing nanoparticles in the small intensine. In my opinion, presented reasearch is interesting and the subject is worth investigation. The article is cosistent and well written and can be accepted for publication in Int. J. Mol. Sci. in the present form.
A1. Thank you very much for your great comments.
Thank you for great comments.

Reviewer 2 Report
An interesting paper of high scientific merit. The topic is very timely and methodologies used appropriate. I have one query which I think may need addressing before publication. These are:
In fig.1 the particle sizes quoted don't seem to match with the cumulative fits shown. Particularly in (C) the 72.1nm particles, the standard deviation looks much greater than 4.2 nm. This also looks odd in Figure 2A. Can the authors please double check and/or address this?
Once this has been addressed I am happy to recommend publication.
Author Response
We carefully revised our manuscript according to the suggestions of the reviewer 2, and details are as follows.
<Q and A for Reviewer 2>
Q1. In fig.1 the particle sizes quoted don't seem to match with the cumulative fits shown. Particularly in (C) the 72.1nm particles, the standard deviation looks much greater than 4.2 nm. This also looks odd in Figure 2A. Can the authors please double check and/or address this?.
A1. The reviewer’s comments are very important. In this study, we used 2 methods (the laser diffraction particle size analyzer SALD-7100 and dynamic light scattering NANOSIGHT LM10) to measure the particle size, and the data from SALD-7100 are expressed as the mean ± standard deviation (S.D.); other data are expressed as the mean ± standard error (S.E.) of the mean. The S.D. in SALD-7100 reflect the particle size distribution (range) (https://www.an.shimadzu.co.jp/powder/lecture/practice/p01/lesson20.htm), and the S.E. in NANOSIGHT LM10 show the deviation of mean particle size in the 5 experiments. In order to respond to the reviewer’s comment, we added the information and URL in the Material and Methods (line 377-380).
Thank you for great comments.

Reviewer 3 Report
In this manuscript, the authors investigated the penetration pathway of indomethacin loaded nanoparticles (IND-NPs) which were developed in their previous work. By adding inhibitors of different endocytosis pathways and controlling the temperature, they found that the absorption of IND-NPs involved energy-dependent endocytosis, and hypothesized that the IND-NPs were firstly taken up by the cells in intestinal epithelium and then dissolved and diffused in the intestine. Overall, the experiments in this work are well organized and most of conclusions are well supported. The results are worth reporting. However, some questions need to be answered and some problems need to be addressed before publication.
1. In the results, the authors claimed: “The drug solubility of the IND-NPs was 131.2 uM”. What does that mean? Could the authors explain it in the results and provide the experimental details in the experimental section?
2. What is the drug loading content and drug loading efficiency of these NPs?
3. Based on the words in results, the Figure 1C is obtained through the analysis of AFM imaging. Could the authors provide the detailed method for that? Is it automated done by the AFM or calculated based on some AFM pictures? Because based on the AFM picture (Figure 1D), the size distribution might be wider than “72.1±4.3 nm”.
4. For the stability tests, could the authors provide some pictures, which will be easier and more visualized, to demonstrate that no precipitation or aggregation after 30 days? See the Figure 3d of this paper “Cellular and Molecular Bioengineering, Volume 9, pp 382-397”.
5. What is the drug release behavior of indomethacin from the NPs? It will be helpful to understand the drug penetration pathway, too.
Author Response
We carefully revised our manuscript according to the suggestions of the reviewer 3, and details are as follows.
<Q and A for Reviewer 3>
Q1. In the results, the authors claimed: “The drug solubility of the IND-NPs was 131.2 uM”. What does that mean? Could the authors explain it in the results and provide the experimental details in the experimental section?
A1. The reviewer’s comment is correct. In this formulations, the indomethacin consisted of 2 type (solution type and solid type), and the ratio of solution/solid type in the IND-MP and IND-NPs were measured by using 25 nm pore size membrane filters (MF™-MEMBRANE FILRER, Merck Millipore, Tokyo, Japan). In order to respond to the reviewer’s comment, we added the information in the Material and Methods, and explained in the Results (line 76-77, 295-297).
Q2. What is the drug loading content and drug loading efficiency of these NPs?
A2. Thank you for pointing out this. The IND-NPs is the indomethacin solid nanoparticles. The IND particle size was decreased using a combination of MC and the bead mill method, and the cohesion of nanoparticulate solids adsorption was prevented by the surface of HPbCD (Ref. 15). The formulations containing solid nanoparticles showed the high bioavailability, and, consequently, an improved therapeutic effect and reduced injury to the small intestine. In addition, we found that the endocytosis related the high membrane penetration of solid nanoparticles in the hydrophobic drugs. In order to respond to the reviewer’s comment, we added the contents in the Materials and Methods (line 288-290).
Q3. Based on the words in results, the Figure 1C is obtained through the analysis of AFM imaging. Could the authors provide the detailed method for that? Is it automated done by the AFM or calculated based on some AFM pictures?.
A3. Thank you very much for pointing this out. Figure 1C is data from measurement by the dynamic light scattering NANOSIGHT LM10. We revised the sentence in the Fig. 1 legend to “…analyzer, and Fig. 1C was measured using the dynamic light scattering method. Figure 1D was observed by SPM-9700” from “…analyzer, and Fig. 1C and 1D were observed by the dynamic light scattering method and SPM-9700, respectively” (line 86-87).
Q4. For the stability tests, could the authors provide some pictures, which will be easier and more visualized, to demonstrate that no precipitation or aggregation after 30 days? See the Figure 3d of this paper “Cellular and Molecular Bioengineering, Volume 9, pp 382-397”.
A4. The reviewer’s comments are very important. In order to respond to the reviewer’s comment, we added the photograph of IND-NPs formulations 30 days after the preparation in the Fig. 2F. (Figure 2F, line 100-102).
Q5. What is the drug release behavior of indomethacin from the NPs? It will be helpful to understand the drug penetration pathway, too.
A5. The IND-NPs is the solid nanoparticles (drug is coated by HPbCD, but is not included), therefore we think that the solid form (size) is important in the penetration. Thank you very much for your great comment.
Thank you for great comments.
